# Protocol describing a systematic review and mixed methods consensus process to define the deteriorated ward patient

James Malycha [ID] ,[1,2,3,4] Chris Andersen,[5] Oliver C Redfern,[3] Sandra Peake,[1,2] Christian Subbe [ID] ,[6] Lukah Dykes,[7] Adam Phillips,[8] Guy Ludbrook [ID] ,[2] Duncan Young,[3] Peter J Watkinson [ID] ,[3] Arthas Flabouris,[2,4] Daryl Jones[9]

**Correspondence to**
Dr James Malycha;
james.malycha@sa.gov.au

## ABSTRACT

**Introduction** Most patients admitted to hospital recover with treatments that can be administered on the general ward. A small but important group deteriorate however and require augmented organ support in areas with increased nursing to patient ratios. In observational studies evaluating this cohort, proxy outcomes such as unplanned intensive care unit admission, cardiac arrest and death are used. These outcome measures introduce subjectivity and variability, which in turn hinders the development and accuracy of the increasing numbers of electronic medical record (EMR) linked digital tools designed to predict clinical deterioration. Here, we describe a protocol for developing a new outcome measure using mixed methods to address these limitations.

**Methods and analysis** We will undertake firstly, a systematic literature review to identify existing generic, syndrome-specific and organ-specific definitions for clinically deteriorated, hospitalised adult patients. Secondly, an international modified Delphi study to generate a short list of candidate definitions. Thirdly, a nominal group technique (NGT) (using a trained facilitator) will take a diverse group of stakeholders through a structured process to generate a consensus definition. The NGT process will be informed by the data generated from the first two stages. The definition(s) for the deteriorated ward patient will be readily extractable from the EMR.

**Ethics and dissemination** This study has ethics approval (reference 16399) from the Central Adelaide Local Health Network Human Research Ethics Committee. Results generated from this study will be disseminated through publication and presentation at national and international scientific meetings.

## STRENGTHS AND LIMITATIONS OF THIS STUDY

⇒ This work addresses an important knowledge gap and will assist in developing electronic, predictive tools for clinical deterioration.
⇒ The systematic review will be thorough and will scope all relevant available published data to inform the development of the definitions.
⇒ The international consensus process will include patients, researchers and clinicians from across different health settings, improving the definition's validity.
⇒ Determining when to implement augmented organ support varies between individual clinicians and healthcare settings and bringing the multiple opinions and experiences together into one set of definitions will be challenging.
⇒ The final definition(s) will be extractable from the electronic medical record and will require ongoing refinement and evaluation in large international data sets.

## INTRODUCTION

Most patients admitted to hospital recover with treatments that can be administered on the general ward. A small but important group deteriorate however, to the extent that they require augmented organ support (figure 1).[1] In observational studies evaluating this cohort, proxy outcomes are used. These include unplanned transfer from the general ward to the intensive care unit (ICU), cardiac arrest and death.[2] The decision to transfer patients to the ICU is dependent on multiple factors, including personalised advance care directives, clinician opinion, local care escalation protocols such as early warning score (EWS) systems, and the availability of ICU resources.[3] Cardiac arrest and death are well defined and easily measured but are often a very late marker of deterioration. Additionally, cardiac arrest frequency is rare, which limits its use for derivation and validation processes, even in large patient data sets. These factors introduce subjectivity and variability to research that uses these as outcome measures, which in turn hinders the development and accuracy of the increasing numbers of electronic medical record (EMR) linked, algorithmic tools designed to predict clinical deterioration.[4 5]

### Aims

The primary aim of this study is to establish an international consensus definition (or set of syndrome or organ-specific definitions) for

BMJ

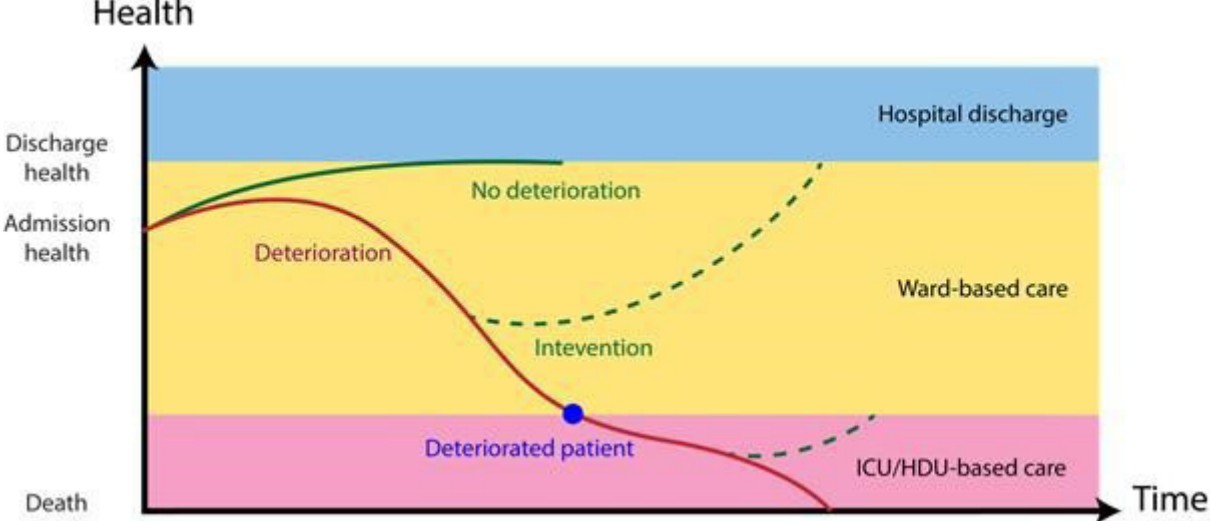

**Figure 1** The schematic representation of potential trajectories for hospitalised patients. Most patients progress along the green line. However, in a small cohort, significant deterioration will occur. This may be subject to early intervention or will reach an end point at which they are no longer suitable for management in a ward environment and will be defined a 'deteriorated'.

the deteriorated ward patient. The secondary aim is to do so using data that is commonly available in most EMRs. The definitions will target the time-point that the requirement for augmented organ support first occurs (while taking into consideration contextual variables such as advanced care directives).

## METHODS
Consensus definitions will be established in three stages. First, a systematic literature review will be undertaken to identify existing generic, syndrome and organ-specific definitions of clinical deterioration. Second, an international modified Delphi study will generate a short list of candidate definitions. Finally, a nominal group technique (NGT) meeting, informed by the data generated from the first two stages, will generate the final definition(s).

### Stage 1: Literature Review
#### Objective
To identify existing generic, syndrome-specific and organ-specific definitions for clinically deteriorated, hospitalised adult patients.

#### Methods
This systematic review will follow the requirements of Preferred Reporting Items for Systematic Reviews and Meta-analyses Protocol.[6]

#### Phenomenon of interest
Studies that characterise or define deteriorated ward patients. These may be regarding generic (or generalised) deteriorated states associated with the traditional EWS systems or novel algorithmic, automated deteriorated patient surveillance tools.[7] They may be regarding syndromes specific to clinical deterioration, such as sepsis and associated definitions including Sepsis-3.[8]

They may also be regarding specific organ dysfunctions, acute decompensated liver failure being a relevant and common example.

### Search strategy
Studies will be identified using Medical Literature Analysis and Retrieval System Online, Excerpta Medica database, Cumulative Index to Nursing and Allied Health Literature, the Cochrane Database of Systematic Reviews and the Cochrane Central Register of Controlled Trials. Additional papers will be sourced from references of included studies, reviews articles and studies from the author libraries.

### Search terms
The following search terms will be included: intensive care, critical care, critical, emergency, deteriorating, deteriorated, definition, electronic patient record, electronic health record, electronic patient record, predictive, unplanned ICU admission, adverse event, terminology, nomenclature, acute, acute care, severe, sudden, rapid response, EWS, sepsis, septic, shock, shocked, hypoxia, COVID-19, respiratory failure, cardiac failure, liver failure, renal failure, anuria, hypotension, instability, unstable, threshold, acute organ dysfunction and criteria. Additional search terms will be considered after initial trials with the above search terms. A trained medical librarian will assist with the search process.

### Study selection and data extraction
Two researchers will independently screen the titles and abstracts of identified studies against the inclusion and exclusion criteria. Disagreement and/or uncertainty regarding study eligibility will be resolved using a third party. Both researchers will independently extract data from included studies using DistillerSR (Evidence

Partners, Ottawa, Canada), which will also be used to manage data and identify duplicates.

## Inclusion criteria
Quantitative or qualitative studies published in peer reviewed journals describing definitions of, or criteria specific to, adult (defined as >16 years of age) clinical deterioration will be eligible for inclusion in this review. Studies published from January 2000 until the day of search completion will be included and no language restrictions will be applied. Google Translate will be used for non-English studies.

## Exclusion criteria
Case studies, editorials, grey literature, letters, practice guidelines and abstract-only reports will be excluded.

## Quality assessment
Different tools will be used to assess the quality of included studies, depending on the type of study. For outcome measure studies, the COnsensus-based Standards for the selection of health Measurement INstruments Risk of Bias checklist will be used.[9] For prediction model studies, the Prediction model Risk Of Bias Assessment Tool will be used.[10] For qualitative studies, the Joanna Briggs Institute Evidence Based Practice Checklist will be used.[11] For clinical guidelines, the Appraisal of Guidelines for Research and Evaluation Instrument (AGREE II) tool will be used.[12] For randomised controlled trials, the Cochrane Collaboration Risk of Bias 2 guidelines will be used.[13]

## Data synthesis and analysis
We will generate a list of organ-specific, syndrome-specific and generic definitions of the deteriorated patient from included studies. These data will be presented in table and text form.

## Stage 2: Delphi Study
### Participants
We aim to include 60 participants in the Delphi study, which is consistent with previous Delphi surveys in critical care outcomes and should be adequate to achieve a suitably diverse international sample of stakeholders.[14] Participants will be recruited through the International Society for Rapid Response Systems, the International Forum for Acute Care Trialists and relevant national societies. No formal inclusion criteria will be used; however, potential participants will be considered based on relevant clinical and research experience, with the aim of ensuring participants are representative of eventual end users. These will include hospital-based clinical staff working in regional, rural and metropolitan hospitals as well as non-clinician content experts such as researchers and digital health specialists.

## Patient and public involvement
A small number of health consumer representatives will also be recruited to participate.

### Round 1: establishing initial definitions (time frame: 2 months)
Results of the literature review and a list of potential domains, variables and/or parameters will be distributed via email to participants. Participants will provide structured feedback on the merits (or otherwise) of each item. These will then be coalesced into an initial list of potential definitions. Any missed items will be submitted to the process for consideration.

### Round 2: ranking potential definitions (time frame: 2 months)
Participants will rank each potential definition using a 9-point Likert System that is recommended by the Grading of Recommendations, Assessment, Development and Evaluation Working Group Handbook for evaluating outcomes measures. Based on previous work in outcomes research, we have defined consensus as 70% of respondents classifying the definitions as 'critical' (score of 7–9) and less than 15% determining the definition to be 'not relevant' (score 1–3). The results will be aggregated. Any criteria achieving a score of >70% 'not relevant' will be removed.

### Round 3: refining aggregated results (time frame: 2 months)
Aggregated results will be presented to each participant. Definitions that remain, but that have not yet achieved consensus, will be rescored. These results will then be aggregated, and the list finalised. Any definitions that have not achieved consensus after three rounds of scoring will be excluded.

### Round 4: generating thresholds (time frame: 2 months)
Participants will propose one threshold for each organ-specific definition with an evidence-based justification for the threshold.

## Stage 3: NGT/consensus meeting (time frame: 1 day)
NGT is a validated method for establishing consensus on a specific issue or range of related issues that uses a trained facilitator to take a group of participants through a structured process.[15 16] The NGT meeting will aim to include a diverse range of clinical stakeholders. The target number of participants will be 15–20.

### Participants
Participants (both professional and public) will be selected and invited using the same process as described for the Delphi. Participants need not have been involved in the first two stages of the study to take part.

A trained facilitator will lead NGT participants through the structured multistage process. First, participants will be presented with an overview of the NGT meeting rationale and aim. Next, participants will be presented with the results of the systematic review and the Delphi process. Participants will then spend 10–15 min writing a list of bullet point reflections and opinions on the definitions provided, including an opportunity to advocate for additional relevant data not previously included. The facilitator will then get participants to list one reflection/opinion that is yet to be presented. Each original

point will be transcribed onto a screen or whiteboard, so all participants can consider and review. This may take several rounds until opinions are exhausted (the aim being to enable all participants to express their views and prevent specific participants having a disproportionate influence).

Participants will then place each definition into two columns: one for inclusion and one for exclusion. The results of this activity will be tabulated and presented. Consensus will be confirmed if more than 70% of participants support its inclusion or exclusion.[16] If there is a lack of consensus on a definition, then the contentious item will be taken back to the group for reappraisal and repeated voting until either consensus or stalemate (two additional voting rounds without consensus) is reached.

The final stage of the NGT will determine the thresholds (if required) for each of the definitions. Participants will write down opinions/reflections on potential thresholds and these will be collated with each original perspective transcribed. Participants will then provide specific thresholds for relevant definitions; these results will be tabulated, and discussion will be encouraged. The facilitators will present numerous potential thresholds based on the feedback and these will again be voted on. The final set of definition thresholds will be presented to the group and pending agreement, recorded for subsequent publication.

### Ethics and dissemination
This study has ethics approval (reference 16399) from the Central Adelaide Local Health Network Human Research Ethics Committee. Results generated from this study will be disseminated through publication and presentation at national and international scientific meetings.

### DISCUSSION
This protocol describes a three-step consensus building process for developing a definition (or set of definitions) for the deteriorated ward patient. The purpose of this research is to aid the development and improve the performance of automated, EMR linked, digital models that predict clinical deterioration in general ward patients. It will also be useful in evaluations of EWS and Rapid Response Systems. It is important to note the definition(s) will not be designed as real-time decision-making adjuncts or to replace complex clinical decision-making.

The work has a number of weaknesses. The pandemic has limited the ability of those involved to gather in person (which is the most effective way to build consensus). The increased familiarity with virtual meeting platforms mitigates this to a certain degree. Determining when to implement augmented organ support varies between individual clinicians and is influenced by institutional resources and healthcare settings. Indeed, defining 'augmented organ support' is itself fraught with difficulty. It is anticipated bringing the multiple opinions and experiences together into

one set of definitions is going to be challenging. Additionally, maintaining consistency across generic, organ-specific (ie, respiratory failure) and syndrome-specific (ie, sepsis) definitions of the deteriorated patient will require discipline and careful planning or risk being of little use in research or clinical practice.

This work has a number of strengths. The specific research question and the methods are novel. The systematic review will be thorough and will ensure all relevant available published data will inform the subsequent modified Delphi survey and NGT, which are the most common and well validated methods for establishing consensus in the medical literature. The definition(s) generated by the study will be evaluated for use as outcome measures when developing predictive tools for clinical deterioration. These may in turn reduce the dependence on the traditional outcome measures, including death, cardiac arrest or unplanned ICU admission, which have specific shortcomings that hinder performance. The definitions will be derived from commonly available EMR data, making them widely applicable as digital healthcare systems become more widespread. There may be additional uses of the consensus definition(s) beyond the remit of this study, such as comparing acuity between different healthcare providers and guiding policy. Overall, the published results will (through various means) be relevant to the many thousands of patients annually who clinically deteriorate on hospital wards.

### Trial status
This is Protocol V.1 dated on 11 February 2022. Recruitment for this study has not begun. It is expected that recruitment for participation in the Delphi and NGT will be completed by 31 December 2022.

**Author affiliations**
[1]Intensive Care Unit, The Queen Elizabeth Hospital, Woodville South, South Australia, Australia
[2]Department of Acute Care Medicine, The University of Adelaide, Adelaide, South Australia, Australia
[3]Kadoorie Centre for Critical Care Research and Education, University of Oxford, Oxford, UK
[4]Intensive Care Unit, Royal Adelaide Hospital, Adelaide, SA, Australia
[5]Critical Care Program, The George Institute for Global Health, Newtown, New South Wales, Australia
[6]School of Medical Sciences, Bangor University, Bangor, UK
[7]Flinders University, Adelaide, South Australia, Australia
[8]University of South Australia, Adelaide, South Australia, Australia
[9]Intensive Care Unit Austin Hospital, Austin Health, Heidelberg, Victoria, Australia

**Contributors** JM and CA conceptualised the study. JM, CA, OCR, CS and DJ undertook the initial methodological planning. GL, PJW, DY, SP, AF undertook secondary analysis of the methods and provided updates and corrections. AP and LD provided technical advice. JM and CA wrote the initial manuscript. OCR, GL, PJW, DY, SP, CS, LD, AP, AF and DJ edited successive drafts and approved the final manuscript.

**Funding** PJW and OCR are both supported by the National Institute for Health and Research Biomedical Research Centre, Oxford. CS does consultancy work for Philips Healthcare.

**Competing interests** None declared.

**Patient and public involvement**  Patients and/or the public were involved in the design, or conduct, or reporting, or dissemination plans of this research. Refer to the Methods section for further details.

**Patient consent for publication**  Not applicable.

**Provenance and peer review**  Not commissioned; externally peer reviewed.

**ORCID iDs**
James Malycha http://orcid.org/0000-0002-9668-1431
Christian Subbe http://orcid.org/0000-0002-3110-8888
Guy Ludbrook http://orcid.org/0000-0001-6925-4277
Peter J Watkinson http://orcid.org/0000-0003-1023-3927

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
