## [Reviewer comments · BMJ Open]

ARTICLE DETAILS

TITLE (PROVISIONAL)	A Protocol Describing A Systematic Review And Mixed Methods Consensus Process To Define The Deteriorated Ward Patient
AUTHORS	Malycha, James; Andersen, Chris; Redfern, Oliver; Peake, Sandra; Subbe, Christian; Dykes, L; Phillips, Adam; Ludbrook, Guy; Young, Duncan; Watkinson, Peter; Flabouris, Arthas; Jones, Daryl

VERSION 1 – REVIEW

REVIEWER	Gunawan, Joko Chulalongkorn University
REVIEW RETURNED	25-Oct-2021

GENERAL COMMENTS	The methods for this study will be redundant. Both Delphi and NGT techniques are a consensus method. The NGT has been used to explore consumer and stakeholder views, while the Delphi technique is commonly used to develop guidelines with health professionals. So, what the authors actually want?
--

REVIEWER	Martín-Conty , José L Universidad de Castilla-La Mancha
REVIEW RETURNED	26-Nov-2021

GENERAL COMMENTS	This research has an original objective and a meaningful content. I congratulate the authors for the work done. I am grateful with the editors for the possibility of revising this manuscript. Although the quality of the manuscript is high, I would like to make some contributions that I hope will increase it and improve readers' understanding. The introduction is clear and well worked. The study design is appropriate and well described, I miss what the keywords were, the search string, the inclusion and exclusion criteria of the study, the discards for not meeting objectives. Discussion is well oriented.
---

REVIEWER	Schenck, Edward Weill Cornell Medical College, Medicine
REVIEW RETURNED	20-Dec-2021

GENERAL COMMENTS	This is a protocol for the development of consensus definitions for the deterioration of ward patients. The study has three planned stages. Overall the topic is of interest and the plan has the potential to improve the critical care communities' approach to patients who experience "worsening" on the ward. The scope is ambitious.
--

	Major comments:  1. The rationale for a consensus definition is clear to those who are clinical researchers in clinical acute medicine. However, more general medical readers may benefit from explicit examples of the increase in predictive modeling in the setting of other forms of organ failure such as sepsis. In that regard, sepsis definitions (including sepsis-3) will likely figure prominently in the systematic review and Delphi process as sepsis events are linked to a plurality of unplanned ICU transfers. Please add details of how your processes will complement sepsis prediction efforts. 2. The investigators seem to understand how much the increase in HER machine learning studies will be important in the consensus definitions but the examples in the search terms are not representative of these features. 3. The introduction/rationale and search terms highlighted for literature review do not give examples of other types of specific organ failure support that may be included in evidence synthesis. There are a plethora of examples of predictive algorithms for deterioration such as acute hypoxemic respiratory failure or disease states such as covid-19 or sepsis as mentioned above. Please consider adding details of how your definitions will complement these evolving literatures. 4. In regard to the literature review, it is unclear how the collected evidence will be synthesized. More concretely, what questions are going to be asked and how will they be quantified for the Delphi processes? Will definitions of organ failure support be tabulated, thresholds, or both? Will the evidence review be formal or semi-structured?  a. Also, who will be performing the literature review? If more than one investigator is performing the review how will the evaluate studies? 5. For the Delphi study, why was the number 60 participants chosen? 6. I am unfamiliar with the 70% threshold for consensus definitions, is this common in the literature. If there is a specific rationale for this choice, please explicitly state it. 7. Please add a sentence describing nominal group techniques, this may not be well known to general readers. 8. The discussion could benefit from lengthening, including adding details regarding potential limitations to the work.
--	---

VERSION 1 – AUTHOR RESPONSE

Reviewer 1 - Dr. Joko Gunawan, Chulalongkorn University

Comments to the Author:

- The methods for this study will be redundant. Both Delphi and NGT techniques are a consensus method. The NGT has been used to explore consumer and stakeholder views, while the Delphi technique is commonly used to develop guidelines with health professionals. So, what do the authors actually want?

We thank the reviewer for taking the time to consider our manuscript. We agree with both comments above and hope that additions outlined below will assist in explaining what we wish to achieve. Namely, to use a three step process to build consensus for a definition of the deteriorated ward patient.

Reviewer 2 - Dr. José L. Martín-Conty, Universidad de Castilla-La Mancha

Comments to the Author:

- I missed what the keywords were, the search string, the inclusion and exclusion criteria of the study, the discards for not meeting objectives.

A lack of detail on the methods of the systematic review were highlighted by the editor and review 3 also. This has now been amended with significant detail added. All the issues raised have been addressed under the section Methods - Stage 1 - Literature Review. We have added paragraphs entitled: Phenomenon of interest, search strategy, search terms, study selection and data extraction, inclusion and exclusion criteria, quality assessment and data synthesis and analysis.

Reviewer 3 - Dr. Edward Schenck, Weill Cornell Medical College

Comments to the Author:

- More general medical readers may benefit from explicit examples of the increase in predictive modelling in the setting of other forms of organ failure such as sepsis.

We hope this point has been adequately addressed in the updated Introduction, Aims, Methods (Systematic Review - Methods - Phenomenon of interest), Discussion.

- In that regard, sepsis definitions (including sepsis-3) will likely figure prominently in the systematic review and Delphi process as sepsis events are linked to a plurality of unplanned ICU transfers.

Please add details of how your processes will complement sepsis prediction efforts.

We agree with Reviewer 3 when they suggest sepsis is a common and important antecedent to ward patients requiring augmented organ support. It is anticipated that elements of the sepsis-3 definitions, such as a vasopressor requirement to maintain mean arterial pressure of <65mmHg, will feature in the final list of definitions for the deteriorated patient. Our intention in writing this protocol is to remain agnostic to likely inclusions in the final criteria to avoid creating bias, real or perceived. Note the added comments in the updated Systematic Review - Methods - Phenomenon of Interest paragraph. And also note additional comments throughout the manuscript with an updated (and repeated) description of the aims being to define generic, syndrome specific and disease specific definitions of the deteriorated patient.

- The investigators seem to understand how much the increase in EHR machine learning studies will be important in the consensus definitions but the examples in the search terms are not representative of these features.

This has been added to the paragraph 'Search Terms' in Methods.

- The introduction/rationale and search terms highlighted for literature review do not give examples of other types of specific organ failure support that may be included in evidence synthesis. There are a plethora of examples of predictive algorithms for deterioration such as acute hypoxemic respiratory failure or disease states such as covid-19 or sepsis as mentioned above. Please consider adding details of how your definitions will complement these evolving literatures.

The search terms have been updated to include a variety of organ specific manifestations of deterioration (including COVID-19)

- In regard to the literature review, it is unclear how the collected evidence will be synthesised. More concretely, what questions are going to be asked and how will they be quantified for the Delphi processes? Will definitions of organ failure support be tabulated, thresholds, or both? Will the evidence review be formal or semi-structured?

The evidence review will be conducted formally and we have re-written the protocol to make this more explicit. The specific questions for the Delphi survey will be derived primarily from the literature review. In general, we will present previously described parameters and thresholds for determining a deteriorated patient that are also frequently recorded in the EMR and request that participants score these parameters from critically important (9) to not important at all (1).

- Also, who will be performing the literature review? If more than one investigator is performing the review how will the evaluate studies?

Please see the updated Systematic Review - Methods paragraphs.

- For the Delphi study, why was the number 60 participants chosen?

There are no clear guidelines for determining the size of a modified Delphi survey. We have chosen 60 as we feel this is likely to enable a suitably diverse range of stakeholders and is consistent with previous work in this area. We have updated the manuscript to explain our rationale.

- I am unfamiliar with the 70% threshold for consensus definitions, is this common in the literature. If there is a specific rationale for this choice, please explicitly state it.

These thresholds were proposed by the Core Outcomes in Medical Effectiveness Trials Handbook (Williamson, 2017) and are now the most commonly applied thresholds for determining consensus around outcome measures.

- Please add a sentence describing nominal group techniques, this may not be well known to general readers.

We have added an additional sentence to the relevant paragraph.

- The discussion could benefit from lengthening, including adding details regarding potential limitations to the work.

We have re-written the discussion including a more detailed description of strengths and weaknesses.

VERSION 2 – REVIEW

REVIEWER	Schenck, Edward Weill Cornell Medical College, Medicine
REVIEW RETURNED	26-Mar-2022
GENERAL COMMENTS	Thank you for responding to my critiques. Overall, my questions are satisfactorily answered. However, the new abstract is now ~500 words and could be significantly shortened.

VERSION 2 – AUTHOR RESPONSE

Dear MBJ Open

Thanks for these suggested minor revisions.

They have been corrected.

Please note the ethics application is currently under review but has not been given before the April 12th deadline for this manuscript.

I will update the system as soon as I get word.

Hopefully this is not delayed as I understand this is a precondition for publication.